# Unsupervised cross domain adaptive anomaly detection network for Internet of Things traffic

Tiange Yuan *, Di Zhai, Anchao Li

Research Institute Of Nuclear Power Operation, China Nuclear Power Operation Technology Corporation, LTD., Wuhan, China

* tiangeyuan@126.com

## Abstract

The rapid growth of networks of connected devices demands robust methods for detecting anomalies in multivariate traffic. Traditional approaches often fail when data distributions shift across environments or when labeled anomalies are scarce. We introduce the Unsupervised Cross Domain Adaptive Anomaly Detection Network called CDA-ADN. This framework employs a conditional variational sequence encoder with temporal attention to learn domain invariant representations of traffic sequences. Domain specific adaptation layers align input and output distributions by applying transformations guided by latent features. A contrastive learning mechanism at both local time steps and global sequence levels separates normal patterns from anomalies. Training occurs in two stages. In the first stage the model learns general normal behavior on a source environment using only normal samples. In the second stage a very small number of unlabeled normal samples from the target environment are used for lightweight fine tuning of the adaptation layers, without requiring any labeled anomaly samples in the target domain. Experiments on two benchmark Internet of Things traffic datasets demonstrate that CDA-ADN outperforms auto encoder and variational auto encoder methods by a wide margin in accuracy Matthews correlation coefficient and sensitivity under label scarce conditions. These results confirm the efficacy of the unsupervised cross domain approach for real world IoT security.

## Introduction

With the increasing deployment of Internet of Things (IoT) devices across industries such as manufacturing, healthcare, and smart infrastructure, the volume of multivariate network traffic data has grown significantly [1]. This data, continuously generated by sensors and interconnected systems, is essential for monitoring system performance, ensuring security, and detecting anomalous behaviors. However, identifying anomalies in IoT traffic remains a challenging task due to the dynamic nature of data, the presence of concept drift, and the scarcity of labeled anomalies [2,3].

**Data availability statement:** All data used in this study are publicly available. The WUSTL-IIoT-2021 dataset is available from the Washington University in St. Louis research repository at https://www.cse.wustl.edu/~jain/iiot2/index.html. The ACI-IoT-2023 dataset is available from IEEE Dataport under DOI: https://doi.org/10.21227/qacj-3x32 and is also accessible via Kaggle at https://www.kaggle.com/datasets/emilynack/aci-iot-network-traffic-dataset-2023. The ToN_IoT dataset is available from the UNSW Canberra research repository at https://research.unsw.edu.au/projects/toniot-datasets.

**Funding:** The author(s) received no specific funding for this work.

**Competing interests:** The authors have declared that no competing interests exist.

Anomaly detection plays a critical role in ensuring the security and reliability of IoT networks by identifying deviations from normal behavior that may indicate cyber-attacks, faults, or system failures [4]. Conventional methods, including rule-based techniques and supervised machine learning models, often rely on predefined thresholds or require extensive labeled datasets. These approaches struggle to generalize across different domains, particularly when data distributions shift due to environmental changes or evolving attack patterns [5]. Moreover, training deep learning models for anomaly detection often necessitates large amounts of labeled data, which are difficult to obtain in real-world scenarios [6–8].

Existing anomaly detection methods for IoT traffic can be broadly categorized into rule-based, statistical, and machine learning-based approaches. Rule-based methods rely on predefined thresholds or signatures to flag anomalies, making them effective for known attack patterns but inadequate for evolving threats. Statistical techniques, such as autoregressive models and Gaussian mixture models, attempt to model normal traffic distributions and detect deviations, but they often struggle with high-dimensional, dynamic IoT data [9]. More recently, deep learning-based methods, including Autoencoders (AEs), Long Short-Term Memory (LSTM) networks, and Transformer-based architectures, have gained traction for their ability to learn complex temporal dependencies in multivariate time-series data [10,11].

While deep learning models have demonstrated strong performance in anomaly detection, they typically require large amounts of labeled data for supervised training [12,13]. However, in real-world IoT environments, obtaining sufficient labeled anomaly samples is challenging due to data privacy concerns, the rarity of certain attack types, and the high cost of manual annotation [14]. Additionally, traditional deep learning models trained on a specific dataset often fail to generalize across different domains, as variations in network traffic patterns and device characteristics lead to performance degradation [15]. These limitations highlight the need for an adaptive and unsupervised approach that can effectively detect anomalies in diverse and dynamically changing IoT environments without relying on labeled target domain data.

In this work, we adopt an unsupervised cross-domain anomaly detection setting where no labeled anomaly samples from the target environment are required. CDA-ADN learns general normal behavioral patterns from a source domain and subsequently adapts to a new target domain using only a very small number of unlabeled normal samples. These normal samples are employed solely to calibrate domain-specific components and do not introduce supervised anomaly information. This setting aligns with widely accepted assumptions in unsupervised domain adaptation and one-class anomaly detection, ensuring that the proposed framework remains fully unsupervised with respect to anomaly identification.

To address the challenges of cross-domain anomaly detection in IoT environments, we propose the **Cross-Domain Adaptive Anomaly Detection Network (CDA-ADN)** – an unsupervised framework that integrates domain adaptation with multi-granularity contrastive learning. Unlike traditional approaches requiring labeled target data, CDA-ADN operates without requiring labeled target domain data, which

is first verified by [16], leveraging a combination of probabilistic latent space modeling, dynamic feature adaptation, and hybrid contrastive learning to achieve robust anomaly detection across diverse IoT environments. Our main contributions are:

- **Conditional Variational Sequence Encoding:** We introduce a GRU-based conditional variational encoder that captures temporal dependencies and learns transferable representations by conditioning on contextual device features. The encoder enforces structured latent distributions through weighted KL divergence and temporal attention, enabling robust generalization across domains.

- **Dynamic Input-Output Adaptation Layers:** We design domain-specific adaptation layers that dynamically align feature distributions using latent-guided affine transformations. These layers mitigate domain shifts by normalizing input and output sequences, ensuring consistent feature representations across source and target domains.

- **Multi-Granularity Contrastive Learning:** We propose a hybrid contrastive learning framework that aligns features at both local (time-step) and global (sequence) levels. Combined with Wasserstein distance-based domain adaptation, this approach maximizes the separation between normal and anomalous patterns while maintaining domain-invariant feature consistency.

- **Two-Stage Optimization for Domain Adaptation:** We further clarify our contribution by emphasizing that the proposed two-stage optimization strategy includes a lightweight fine-tuning step using only a minimal set of normal target-domain samples. This fine-tuning does not rely on any labeled anomalies and is performed exclusively to adapt the latent-guided affine transformation layers to the target distribution. As such, the method preserves the unsupervised nature of the framework while enabling effective cross-domain alignment in label-scarce Internet of Things environments.

The rest of this paper is organized into four unnumbered sections. Related Work reviews prior approaches to IoT anomaly detection and domain adaptation. Method introduces the CDA-ADN framework, detailing its architecture, adaptation mechanisms, and training strategy. Experiments presents our evaluation results and comparisons with baseline methods. Finally, Conclusion summarizes the main findings and outlines directions for future research.

## Related work

Anomaly detection in multivariate IoT traffic has been extensively studied, evolving from traditional rule-based methods to deep learning-based techniques and transfer learning approaches. Early methods primarily relied on predefined rules and statistical models. Rule-based approaches, such as Snort and Suricata, use known attack signatures or manually set thresholds to detect anomalies [4]. While effective for well-defined attack patterns, these methods fail to detect novel or evolving threats. Statistical models, including autoregressive integrated moving average (ARIMA) [2] and Gaussian mixture models (GMM) [5], attempt to learn normal network behaviors and flag deviations. However, these methods assume stationarity in data distribution and struggle with high-dimensional, dynamic IoT traffic, where concept drift is common [3].

Deep learning techniques have significantly improved anomaly detection by automatically extracting relevant features from raw traffic data [17,18]. Autoencoders and Long Short-Term Memory networks are widely used for time-series anomaly detection, where reconstruction errors indicate deviations from normal patterns [9]. Variational Autoencoders extend AEs by modeling a probabilistic latent space, enhancing robustness to noise and unseen anomalies [6]. More recently, Transformer-based architectures have been introduced to capture long-range dependencies in IoT data, with models such as the Anomaly Transformer [14] leveraging attention mechanisms to distinguish normal from anomalous behaviors. Hybrid models that combine CNNs with LSTMs have also been proposed to integrate spatial and temporal feature learning, achieving promising results on real-world datasets [11]. Despite their advantages, most deep learning methods require large-scale labeled training data, which is difficult to obtain in real-world IoT applications due to the rarity of labeled anomalies and privacy constraints. In parallel, recent surveys have provided comprehensive overviews of deep

learning methods for time-series anomaly detection and graph neural networks for time-series analysis, including anomaly detection tasks [19,20]. These works highlight the rapid development of representation learning techniques for time-series anomalies and motivate the need for models that remain robust under distribution shifts across domains.

To mitigate the reliance on labeled data, transfer learning has been explored as a solution for cross-domain anomaly detection. By leveraging knowledge from a well-defined source domain, transfer learning enables adaptation to a target domain with minimal labeled data. Recent studies have applied domain adaptation techniques using adversarial training and feature alignment to bridge distributional differences [15]. More recent work on time-series domain adaptation under feature and label shifts, as well as sensor-level inter-domain alignment for multivariate time series, further emphasizes the importance of learning domain-invariant temporal representations in an unsupervised or label-efficient manner [21,22]. In the context of IoT security, ResADM [16] demonstrates the potential of transfer learning for cyber-physical systems by improving generalization across datasets.

While existing approaches have made notable progress, they are often constrained by their dependence on labeled target data or their inability to generalize across dynamically changing IoT environments. To address these limitations, we propose the Cross-Domain Adaptive Anomaly Detection Network, an unsupervised transfer learning framework that eliminates the need for labeled target domain data while enhancing adaptability across different IoT environments. By integrating contrastive learning with a GRU-based Conditional Variational Autoencoder, CDA-ADN improves feature discrimination between normal and anomalous patterns. Additionally, input-output adaptation layers mitigate domain discrepancies, allowing the model to generalize effectively to unseen domains. Our approach enables few-shot domain adaptation using only a minimal set of normal target domain samples, reducing computational overhead while maintaining high anomaly detection performance. Experimental results on real-world IoT datasets demonstrate that CDA-ADN outperforms existing methods in terms of accuracy, MCC, and sensitivity, making it a promising solution for scalable and adaptive anomaly detection.

## Method

We propose the CDA-ADN, an unsupervised framework that integrates domain adaptation with multi-granularity contrastive learning to tackle the challenges of detecting anomalies in multivariate IoT traffic. As shown in Fig 1, CDA-ADN

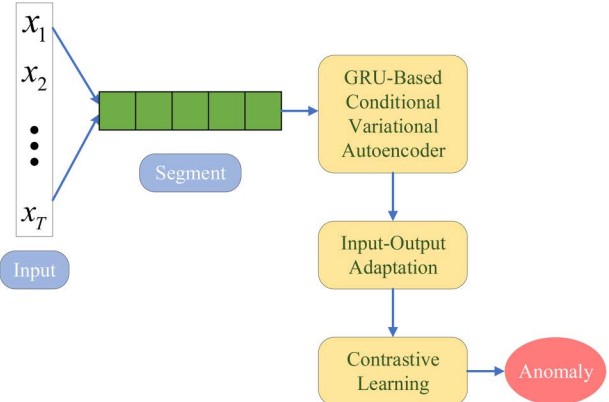

**Fig 1. Framework of the Cross-Domain Adaptive Anomaly Detection Network (CDA-ADN).** The process begins with multivariate IoT traffic data $(\mathbf{X}_1, \mathbf{X}_2, \ldots, \mathbf{X}_T)$ as input. The data is segmented into fixed-length sequences before being processed by a GRU-based Conditional Variational Autoencoder to extract latent representations while preserving temporal dependencies. Input-output adaptation layers align feature distributions between the source and target domains, reducing domain discrepancies. Contrastive learning further enhances feature separation, enabling robust anomaly detection based on the learned representations. Anomalies are identified using a threshold on reconstruction error or feature deviations.

 

features: (1) a conditional variational sequence encoder with temporal attention for cross-domain feature extraction, (2) dynamic input-output adaptation layers for domain alignment, and (3) a multi-granularity contrastive learning mechanism enhanced with Wasserstein distance-based domain adaptation. The framework learns domain-invariant temporal patterns through a variational encoder-decoder, aligns feature distributions via latent-guided normalization, and enforces consistency across domains at local and global levels using hybrid contrastive learning. Unlike conventional methods requiring labeled target data, CDA-ADN achieves fully unsupervised anomaly detection through latent space regularization with probabilistic constraints, minimal target domain fine-tuning, and contrastive alignment, ensuring adaptability to diverse IoT environments.

At the beginning of the training pipeline, CDA-ADN performs source-domain learning using only normal traffic sequences, without requiring any labels from the target domain. During the subsequent domain adaptation stage, a very small number of unlabeled normal target-domain samples are used exclusively to calibrate the domain-specific transformation layers. No labeled anomaly data from the target domain are employed at any point. This refinement ensures terminological consistency and clearly distinguishes our unsupervised setting from semi-supervised alternatives.

## Problem formulation for cross-domain anomaly detection

Anomaly detection in multivariate IoT traffic presents significant challenges due to dynamic variations in network conditions, device heterogeneity, and the absence of labeled anomalies in the target domain. Traditional supervised models struggle to generalize across domains because of distribution shifts, making them ineffective in real-world deployments. To address this issue, we formulate anomaly detection as an unsupervised cross-domain adaptation problem, where the goal is to detect anomalies in an unseen target domain without requiring labeled target data.

We consider a source domain $\mathcal{D}_s$ and a target domain $\mathcal{D}_t$, where each domain consists of multivariate time-series sequences:

$$\mathbf{X} = \{\mathbf{x}_1, \mathbf{x}_2, \ldots, \mathbf{x}_T\}, \quad \mathbf{x}_t \in \mathbb{R}^d,$$

where $T$ represents the sequence length, and $d$ denotes the number of features, such as network traffic attributes or sensor readings. The objective is to learn a function $F_\theta : \mathbb{R}^{T \times d} \to \mathcal{H}$ that transforms sequences into a latent space $\mathbf{h} \in \mathcal{H}$, where normal and anomalous behaviors are well-separated.

However, due to domain shifts, the probability distributions of the source and target domains may differ:

$$P_s(\mathbf{X}) \neq P_t(\mathbf{X}).$$

This distributional discrepancy hinders the direct application of models trained on $\mathcal{D}_s$ to $\mathcal{D}_t$. To overcome this limitation, we incorporate input-output adaptation layers to standardize feature representations and minimize domain discrepancies. Additionally, we leverage contrastive learning to enforce similarity among normal samples and maximize separation between normal and anomalous samples.

Anomalies are identified through reconstruction errors and deviations within the latent space. Given an encoded representation $\mathbf{h}$, the model reconstructs the input sequence $\hat{\mathbf{X}}$, and the reconstruction loss is computed as

$$\mathcal{L}_{rec} = \frac{1}{T} \sum_{t=1}^{T} \|\mathbf{x}_t - \hat{\mathbf{x}}_t\|^2.$$

Higher reconstruction errors indicate deviations from normal traffic patterns, making this a key criterion for anomaly detection. However, relying solely on reconstruction errors may not be sufficient, as certain anomalies may still be reconstructed

with low error. To further improve feature discrimination, we introduce multi-granularity contrastive learning, where representations of similar samples are pulled together while dissimilar ones are pushed apart:

$$\mathcal{L}_{ctr} = \sum_{(i,j)\in\mathcal{P}} \log\sigma(-d(\mathbf{h}_i, \mathbf{h}_j)) + \sum_{(i,j)\in\mathcal{N}} \log\sigma(d(\mathbf{h}_i, \mathbf{h}_j) - m),$$

where $\mathcal{P}$ and $\mathcal{N}$ denote the sets of positive (similar) and negative (dissimilar) pairs, $d(\mathbf{h}_i, \mathbf{h}_j)$ represents the distance between representations, and $m$ is a margin parameter.

To enforce cross-domain feature alignment, we introduce a Wasserstein distance-based domain adaptation loss:

$$\mathcal{L}_{wd} = \inf_{\gamma\in\Gamma(P_s, P_t)} \mathbb{E}_{(\mathbf{h}_s, \mathbf{h}_t)\sim\gamma}[\|\mathbf{h}_s - \mathbf{h}_t\|],$$

where $\Gamma(P_s, P_t)$ denotes the set of all joint distributions with marginals $P_s$ and $P_t$. This loss minimizes distributional discrepancies between $\mathcal{D}_s$ and $\mathcal{D}_t$, enabling better generalization in the target domain.

In the problem formulation, we explicitly state that the few-shot fine-tuning mechanism operates only on a limited subset of unlabeled normal samples from the target domain. This process adjusts the latent-guided affine adaptation layers to account for domain-specific characteristics while preserving the unsupervised nature of anomaly detection, as no anomaly labels from the target domain are used.

The final optimization objective integrates reconstruction loss, contrastive loss, KL divergence, and domain adaptation loss:

$$\mathcal{L} = \lambda_{rec}\mathcal{L}_{rec} + \lambda_{kl}\mathcal{L}_{kl} + \lambda_{ctr}\mathcal{L}_{ctr} + \lambda_{wd}\mathcal{L}_{wd}.$$

Through joint optimization of these components, the model learns domain-invariant representations, ensuring consistency of normal patterns across domains while effectively distinguishing anomalies. This formulation enables unsupervised anomaly detection in the target domain without requiring labeled anomaly samples.

## Conditional variational sequence encoding

To effectively capture temporal dependencies and learn transferable representations, we employ a conditional variational sequence encoder based on a Gated Recurrent Unit (GRU). This encoder transforms an input sequence $\mathbf{X}$ and contextual information $\mathbf{c}_{ctx}$ (e.g., device type, protocol) into a structured latent space where normal and anomalous behaviors can be effectively distinguished. Unlike deterministic autoencoders, our model introduces a probabilistic latent space conditioned on $\mathbf{c}_{ctx}$, enhancing generalization and anomaly detection.

Concretely, the encoder uses a single layer GRU with hidden size $H = 128$ that processes the input sequence $\mathbf{X}$ together with the contextual vector $\mathbf{c}_{ctx}$ and produces a final hidden state $\mathbf{h}_T \in \mathbb{R}^H$. Two linear projection heads $f_\psi$ and $g_\psi$ map $\mathbf{h}_T$ to the mean and variance of a $d_c = 64$ dimensional latent variable $\mathbf{c}$, which defines the conditional posterior $q_\psi(\mathbf{c} \mid \mathbf{X}, \mathbf{c}_{ctx})$ used in the reparameterization step. The KL divergence term is weighted by a coefficient $\lambda_{kl}$ that is linearly increased from 0 to $10^{-3}$ during the first 20 training epochs and then kept fixed, which stabilizes early optimization while still enforcing a well structured latent space at convergence.

Given an input sequence $\mathbf{X} = \{\mathbf{x}_1, \mathbf{x}_2, \dots, \mathbf{x}_T\}$ and contextual information $\mathbf{c}_{ctx}$, the encoder extracts a latent representation by learning the parameters of a variational distribution:

$$q_\psi(\mathbf{c}|\mathbf{X}, \mathbf{c}_{ctx}) = \mathcal{N}\left(\mathbf{c}; f_\psi(\mathbf{X}, \mathbf{c}_{ctx}), \mathrm{diag}(g_\psi(\mathbf{X}, \mathbf{c}_{ctx}))\right),$$

where $f_\psi$ and $g_\psi$ are neural networks that parameterize the mean and variance of the latent distribution. The latent variable $\mathbf{c}$ is sampled as:

$$\mathbf{c} = f_\psi(\mathbf{X}, \mathbf{c}_{\text{ctx}}) + \text{SoftPlus}(g_\psi(\mathbf{X}, \mathbf{c}_{\text{ctx}})) \odot \epsilon, \quad \epsilon \sim \mathcal{N}(\mathbf{0}, \mathbf{I}),$$

where $\text{SoftPlus}(x) = \log(1 + e^x)$ ensures non-negative variance.

To dynamically prioritize critical time steps in the latent space, we introduce a temporal attention mechanism that directly operates on the latent variable $\mathbf{c}$. For each time step $t$, the attention weights $\alpha_t$ are computed as:

$$\alpha_t = \text{Softmax}\left(\mathbf{W}_a \mathbf{c}_t + \mathbf{b}_a\right),$$

where $\mathbf{W}_a$ and $\mathbf{b}_a$ are learnable parameters. The attention-weighted latent representation $\tilde{\mathbf{c}}_t$ is then given by:

$$\tilde{\mathbf{c}}_t = \alpha_t \odot \mathbf{c}_t.$$

This mechanism allows the model to focus on latent features from key time steps, enhancing anomaly detection robustness.

The decoder reconstructs the input sequence from the attention-weighted latent variables $\tilde{\mathbf{c}}_t$ and contextual information $\mathbf{c}_{\text{ctx}}$. The reconstruction is formulated as:

$$\hat{\mathbf{x}}_t = f_{\text{decoder}}(\tilde{\mathbf{c}}_t, \mathbf{c}_{\text{ctx}}),$$

where $f_{\text{decoder}}$ is a GRU-based network. The reconstruction loss is computed using a weighted mean squared error:

$$\mathcal{L}_{rec} = \frac{1}{T} \sum_{t=1}^{T} w_t \|\mathbf{x}_t - \hat{\mathbf{x}}_t\|^2,$$

with $w_t$ being a time-dependent weight function prioritizing critical time steps.

To enforce structured latent distributions while incorporating attention, we propose a weighted KL divergence loss:

$$\mathcal{L}_{kl} = \sum_{t=1}^{T} \alpha_t \cdot D_{KL}\left(q_\psi(\mathbf{c}_t|\mathbf{X}, \mathbf{c}_{\text{ctx}}) \| \mathcal{N}(\mu_0, \text{diag}(\sigma_0^2))\right),$$

where $\alpha_t$ are the attention weights. This ensures that the model pays higher attention to time steps with significant distribution shifts.

## Dynamic domain adaptation layers

In real-world IoT deployments, variations in device configurations, network conditions, and environmental factors introduce domain shifts between the source domain $\mathcal{D}_s$ and the target domain $\mathcal{D}_t$. Directly applying a model trained on $\mathcal{D}_s$ to $\mathcal{D}_t$ often leads to performance degradation due to differences in feature distributions. To mitigate this issue, we introduce input-output adaptation layers that dynamically align feature representations across domains while preserving temporal dependencies. These layers leverage the latent variable $\mathbf{c}$ to guide domain-specific feature transformations, ensuring robust adaptation.

The input adaptation layer standardizes feature distributions by transforming input sequences before passing them into the variational sequence encoder. Concretely, given the time-dependent latent variables $\{\mathbf{c}_t\}_{t=1}^{T}$, we first aggregate them into a sequence-level descriptor

$$\bar{\mathbf{c}} = \frac{1}{T} \sum_{t=1}^{T} \mathbf{c}_t.$$

For each domain $d \in \{s, t\}$, the input adaptation layer predicts feature-wise scaling and shifting vectors from $\bar{\mathbf{c}}$

$$\gamma_d(\bar{\mathbf{c}}) = \mathbf{W}_d^\gamma \bar{\mathbf{c}} + \mathbf{b}_d^\gamma, \qquad \beta_d(\bar{\mathbf{c}}) = \mathbf{W}_d^\beta \bar{\mathbf{c}} + \mathbf{b}_d^\beta,$$

and the adapted input sequence for domain $d$ is computed as

$$\tilde{\mathbf{x}}_t^{(d)} = \gamma_d(\bar{\mathbf{c}}) \odot \mathbf{x}_t^{(d)} + \beta_d(\bar{\mathbf{c}}), \quad t = 1, \dots, T, \ d \in \{s, t\}.$$

This latent-guided affine transformation enables feature-wise alignment of input distributions across domains prior to encoding.

On the output side, the decoder produces reconstructed sequences $\hat{\mathbf{X}}$, which are further refined using an output adaptation layer. Similarly, the output adaptation layer uses $\bar{\mathbf{c}}$ to parameterize domain-specific normalization statistics

$$\mu_d(\bar{\mathbf{c}}) = \mathbf{W}_d^\mu \bar{\mathbf{c}} + \mathbf{b}_d^\mu, \qquad \sigma_d(\bar{\mathbf{c}}) = \text{SoftPlus}\left(\mathbf{W}_d^\sigma \bar{\mathbf{c}} + \mathbf{b}_d^\sigma\right),$$

and aligns reconstructed features across domains via

$$\hat{\mathbf{x}}_t^{(d)} = \frac{\hat{\mathbf{x}}_t - \mu_d(\bar{\mathbf{c}})}{\sigma_d(\bar{\mathbf{c}})}, \quad t = 1, \dots, T, \ d \in \{s, t\}.$$

These latent-guided affine transformations explicitly couple the shared latent representation with domain-wise feature normalization, making the dynamics of the input-output adaptation layers and the resulting cross-domain alignment more transparent and reproducible.

By incorporating these adaptation layers, the model effectively reduces domain discrepancies while enhancing the robustness of feature representations across diverse IoT environments. The use of $\bar{\mathbf{c}}$ as a conditioning variable ensures that the adaptation process is guided by the latent representation of the input sequence, enabling more precise domain alignment.

**Multi-granularity contrastive learning and domain adaptation**

To ensure that the learned feature representations generalize well across domains, we introduce a multi-granularity contrastive learning framework that enhances domain-invariant representation learning. The objective is to encourage feature consistency at both local (time-step) and global (sequence) levels while maximizing the separation between normal and anomalous patterns.

Let $\mathbf{c}_s$ and $\mathbf{c}_t$ denote the latent representations of sequences from the source domain $\mathcal{D}_s$ and target domain $\mathcal{D}_t$, respectively. Given a batch of $N$ sequences, we construct positive and negative pairs at two granularities:

• Time-Step Level: For each sequence, positive pairs are formed between adjacent time steps, while negative pairs are formed between distant time steps.

• Sequence Level: Positive pairs are formed between sequences of the same class (normal or anomalous), while negative pairs are formed between sequences of different classes.

Formally, the positive and negative sets are defined as:

$$\mathcal{P}_{\text{local}} = \{(\mathbf{c}_t, \mathbf{c}_{t+1})\}, \quad \mathcal{N}_{\text{local}} = \{(\mathbf{c}_t, \mathbf{c}_{t+k}) \mid k > 1\}$$

$$\mathcal{P}_{\text{global}} = \{(\mathbf{c}_i, \mathbf{c}_j) \mid y_i = y_j\}, \quad \mathcal{N}_{\text{global}} = \{(\mathbf{c}_i, \mathbf{c}_j) \mid y_i \neq y_j\}$$

where $y_t$ in the target domain is estimated based on feature proximity to known normal samples.

The contrastive loss integrates both local and global granularities:

$$\mathcal{L}_{ctr} = \alpha\mathcal{L}_{\text{local}} + (1-\alpha)\mathcal{L}_{\text{global}}$$

where $\alpha$ is a balancing hyperparameter. Both local and global losses share the same formulation:

$$\mathcal{L} = \sum_{(i,j)\in\mathcal{P}} \log\sigma(-d_{ij}) + \sum_{(i,j)\in\mathcal{N}} \log\sigma(d_{ij}-m)$$

where $\mathcal{P}, \mathcal{N}$ correspond to local or global positive and negative pairs, $d_{ij} = d(\mathbf{c}_i, \mathbf{c}_j)$ denotes the distance between latent representations, $\sigma(\cdot)$ is the sigmoid function, and $m$ is a margin hyperparameter.

To align feature distributions across domains, we use the Wasserstein distance-based domain adaptation loss:

$$\mathcal{L}_{wd} = \inf_{\gamma\in\Gamma(P_s,P_t)} \mathbb{E}_{(\mathbf{c}_s,\mathbf{c}_t)\sim\gamma}[\|\mathbf{c}_s - \mathbf{c}_t\|],$$

where $\Gamma(P_s, P_t)$ denotes the set of all joint distributions with marginals $P_s$ and $P_t$.

We choose the Wasserstein-1 distance for domain alignment because it provides a well-behaved discrepancy measure with informative gradients even when the source and target feature distributions have limited overlap under large domain shifts. In contrast to the Jensen–Shannon divergence commonly used in GAN-based alignment, which may lead to vanishing gradients when the two distributions are far apart, Wasserstein-1 directly reflects the optimal transport cost required to move probability mass from the source latent space to the target latent space, which typically yields more stable optimization. This property is particularly important in our setting, where the latent representations learned from heterogeneous IoT environments may exhibit substantial mismatch. Following the standard formulation of Wasserstein learning, we implement the objective with a gradient-penalized critic network to enforce the Lipschitz constraint, which further stabilizes training, improves reproducibility, and ultimately yields more reliable cross-domain alignment in the latent space [23,24].

## Overall training objective

To ensure clarity and reproducibility, we summarize all loss components of CDA-ADN in a unified formulation. The domain-adaptation loss is implemented using the Wasserstein-1 distance with a gradient-penalized critic network. Given source-domain latent representations $z_s$ and target-domain latent representations $z_t$, the critic $D(\cdot)$ is trained to maximize

$$\mathcal{L}_W = \mathbb{E}_{z_s}[D(z_s)] - \mathbb{E}_{z_t}[D(z_t)],$$

while enforcing 1-Lipschitz continuity via the gradient penalty

$$\mathcal{L}_{GP} = \lambda\,\mathbb{E}_{\hat{z}}\left(\|\nabla_{\hat{z}}D(\hat{z})\|_2 - 1\right)^2,$$

where $\hat{z}$ is sampled along straight lines between $z_s$ and $z_t$. The domain-adaptation objective is therefore

$$\mathcal{L}_{DA} = \mathcal{L}_W + \mathcal{L}_{GP}.$$

For contrastive learning, we employ margin-based losses at both the local time-step level and the global sequence level. For a pair of latent features $(h_i, h_j)$ with pseudo-label $y_{ij} \in \{0, 1\}$ indicating positive or negative relation, the contrastive loss is

$$\mathcal{L}_{\text{con}}(h_i, h_j) = y_{ij}\|h_i - h_j\|_2^2 + (1-y_{ij})\max(0, m - \|h_i - h_j\|_2)^2,$$

where $m$ is the contrastive margin. Local and global contrastive objectives are combined as

$$\mathcal{L}_{CL} = \alpha\mathcal{L}_{local} + (1 - \alpha)\mathcal{L}_{global},$$

with balance coefficient $\alpha$ and temperature parameter $\tau$ used in the similarity normalization process.

Finally, the total training objective integrates the variational reconstruction loss $\mathcal{L}_{VAE}$, the domain-adaptation loss $\mathcal{L}_{DA}$, and the multi-granularity contrastive loss $\mathcal{L}_{CL}$:

$$\mathcal{L}_{total} = \mathcal{L}_{VAE} + \beta\mathcal{L}_{DA} + \gamma\mathcal{L}_{CL},$$

where $\beta$ and $\gamma$ are weighting coefficients. Table 5 provides the key hyperparameters used in the contrastive and domain-adaptation losses for completeness.

## Training and optimization strategy

The final objective of our model is to jointly optimize reconstruction accuracy, latent space regularization, multi-granularity contrastive learning, and domain-invariant feature alignment. The total loss function integrates the reconstruction loss $\mathcal{L}_{rec}$, the KL divergence loss $\mathcal{L}_{kl}$, the multi-granularity contrastive loss $\mathcal{L}_{ctr}$, and the Wasserstein distance-based domain adaptation loss $\mathcal{L}_{wd}$:

$$\mathcal{L} = \lambda_{rec}\mathcal{L}_{rec} + \lambda_{kl}\mathcal{L}_{kl} + \lambda_{ctr}\mathcal{L}_{ctr} + \lambda_{wd}\mathcal{L}_{wd},$$

where $\lambda_{rec}$, $\lambda_{kl}$, $\lambda_{ctr}$, and $\lambda_{wd}$ are weighting coefficients that balance different loss components.

To train the model, we employ a two-stage optimization process, as illustrated in Fig 2. In the first stage, the conditional variational sequence encoder and decoder are pre-trained on the source domain $\mathcal{D}_s$ to learn normal behavior representations by minimizing:

$$\mathcal{L}_{pretrain} = \lambda_{rec}\mathcal{L}_{rec} + \lambda_{kl}\mathcal{L}_{kl}.$$

This step ensures that the model learns a structured latent space before applying cross-domain adaptation.

In the second stage, the model is fine-tuned using both $\mathcal{D}_s$ and $\mathcal{D}_t$, incorporating multi-granularity contrastive learning and Wasserstein distance-based domain adaptation. The contrastive pairs for $\mathcal{L}_{ctr}$ are dynamically generated using a memory bank approach to maintain representative samples and enhance feature discrimination across domains.

The above two-stage procedure assumes access to a small subset of mostly normal samples from the target domain that are used to calibrate the latent-guided adaptation layers. When this subset is extremely small, the adaptation parameters become underconstrained and the benefit over source-only training is reduced, so CDA-ADN behaves closer to a source-only detector that still leverages contrastive and Wasserstein regularization but with limited domain-specific refinement. In practice, the target subset may also contain a small fraction of anomalous samples. Since these samples are treated as normal during fine-tuning, they can bias the adapted feature statistics and slightly reduce sensitivity to those particular anomalous patterns. A natural extension is to combine the adaptation step with a simple robustness mechanism, for example by discarding target samples with very high reconstruction error before updating the adaptation layers or by using robust aggregation operators instead of plain averages when estimating target-domain statistics. Such strategies preserve the unsupervised nature of CDA-ADN while mitigating the impact of scarce or mildly contaminated target samples.

The optimization follows the Adam optimizer with a learning rate $\eta$:

$$\theta^* = \arg\min_{\theta} \mathbb{E}[\mathcal{L}(\theta)],$$

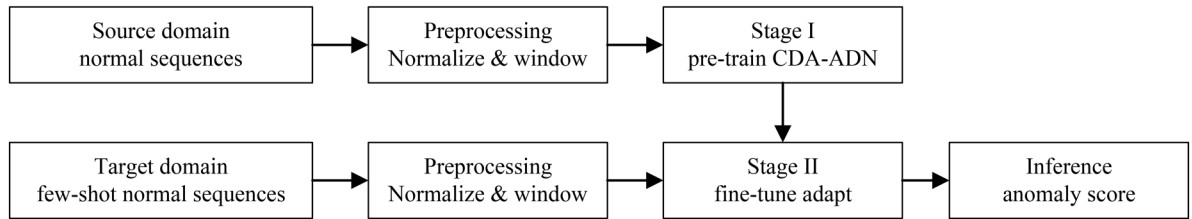

**Fig 2. Two-stage training workflow of CDA-ADN.** Stage I pre-trains the model on source-domain normal traffic to learn general normal representations. Stage II performs lightweight adaptation on the target domain using only a small subset of normal samples, where only the domain-specific adaptation components are fine-tuned while the shared encoder-decoder is kept fixed. After adaptation, the model is directly applied to the target domain to compute anomaly scores and make anomaly decisions.

where $\theta$ represents the model parameters.

To ensure stable training, we apply batch normalization within the input-output adaptation layers and employ early stopping based on validation loss to prevent overfitting. The final trained model is evaluated on the target domain $\mathcal{D}_t$ without requiring labeled anomaly samples, demonstrating its effectiveness in unsupervised cross-domain anomaly detection.

## Experiments

To evaluate the effectiveness of the proposed method, we conduct experiments on real-world IoT datasets. The evaluation focuses on comparing CDA-ADN with baseline methods, analyzing the impact of key components such as contrastive learning and adaptation layers, and assessing the model's robustness and consistency. Results demonstrate the superior cross-domain anomaly detection performance of CDA-ADN, showcasing its potential for IoT applications.

### Experimental setup

The effectiveness of CDA-ADN is evaluated on two real-world IoT datasets. The performance is compared with baseline methods using multiple evaluation metrics to assess both classification accuracy and domain generalization.

The first dataset, WUSTL-IIOT-2021 [25], consists of network traffic collected from an industrial IoT system. It includes both normal traffic and attack scenarios such as denial-of-service (DoS), injection, and backdoor attacks. The second dataset, ACI-IoT-2023 [26], is collected from a real-world IoT environment and contains diverse cyber threats, including reconnaissance, brute-force attacks, and distributed denial-of-service (DDoS). These datasets differ significantly in terms of network structure and attack patterns, making them suitable for evaluating the cross-domain adaptability of the proposed model. In addition, we include the ToN_IoT dataset [27], a large-scale IoT telemetry and network traffic benchmark containing both benign behaviors and diverse cyberattacks. Its heterogeneous sources and attack diversity make it a challenging target domain for evaluating cross-domain anomaly detection under distribution shifts.

To quantify model performance, we use accuracy, Matthews correlation coefficient (MCC), and sensitivity. Accuracy measures the proportion of correctly classified instances, while MCC provides a balanced evaluation considering true positives, false positives, and false negatives, making it well-suited for imbalanced anomaly detection tasks [28]. Sensitivity reflects the model's ability to correctly identify anomalies, which is critical for security applications.

CDA-ADN is implemented using PyTorch and trained on an NVIDIA A100 GPU. The model is optimized using the Adam optimizer [29] with an initial learning rate of $10^{-3}$ and a batch size of 128. Training is performed for a maximum of 100 epochs with early stopping based on validation loss. The contrastive loss margin is set to 0.5, while the weights for the domain adaptation and reconstruction losses are set to 0.1 and 1.0, respectively.For both datasets, continuous traffic features are standardized using z-score normalization with mean and standard deviation computed on the source-domain training split and then reused for all target-domain samples. The normalized streams are segmented into fixed-length

windows of $T=128$ time steps with a stride of $S=64$, which serve as the multivariate inputs to CDA-ADN. The main training configurations are summarized in Table 1.

Anomalies are decided by thresholding a reconstruction based anomaly score. The decision threshold $\tau$ is selected on a held out validation subset that contains only normal samples from the source domain by setting $\tau$ to the 95th percentile of the validation scores.

## Baseline methods

To evaluate the performance of CDA-ADN, we compare it with state-of-the-art anomaly detection models, including both conventional statistical models and deep learning-based approaches.

- **Autoencoder (AE) [30]**: An unsupervised deep learning model that reconstructs input data and detects anomalies based on reconstruction errors.

- **Variational Autoencoder (VAE) [6]**: A probabilistic extension of AE that learns a structured latent space, improving anomaly detection generalization.

- **Anomaly Transformer (AT) [14]**: A Transformer-based time-series anomaly detector that models long-range temporal dependencies and measures association discrepancy between time steps to highlight abnormal patterns.

- **Graph Deviation Network (GDN) [31]**: A graph neural network based model that leverages an adaptive sensor graph to capture feature correlations and detects anomalies by measuring deviations from learned graph-structured normal patterns.

Both AE and VAE are trained on normal data and rely on reconstruction errors for anomaly detection. These methods serve as strong baselines for evaluating CDA-ADN's ability to enhance anomaly separation and domain adaptation.

## Experimental results and analysis

The performance of CDA-ADN and baseline methods on the target domain is summarized in Tables 2 and 3. To provide a more comprehensive evaluation, we report the mean and standard deviation across multiple runs for five metrics: Accuracy, MCC, Sensitivity, AUC-ROC, and AUC-PR. While Accuracy, MCC, and Sensitivity reflect thresholded classification performance, AUC-ROC and AUC-PR offer threshold-independent views that are particularly informative under class imbalance.

Tables 2 and 3 summarize the cross-domain detection results under two transfer settings, i.e., WUSTL-IIoT-2021→ACI-IoT-2023 and WUSTL-IIoT-2021 → ToN_IoT. Across both target domains, CDA-ADN consistently achieves the best overall performance against all baselines, including reconstruction-based methods (AE, VAE), Transformer-based AT, and graph-based GDN, indicating strong robustness under domain shift.

**Table 1. Training setup summary.**

| Parameter | Value |
| --- | --- |
| Optimizer | Adam |
| Learning rate | $1 \times 10^{-3}$ |
| Batch size | 128 |
| Epochs | 100 |
| Random seeds | 2025 |
| Number of trials | 5 |
| Hardware | NVIDIA A100 |

**Table 2. Cross-domain detection performance from WUSTL-IIoT-2021 (source) to ACI-IoT-2023 (target). Values are reported as mean±standard deviation over five independent runs.**

| Model | Acc (%) | MCC | Sen | Spe | F1 | AUC-ROC | AUC-PR |
|---|---|---|---|---|---|---|---|
| AE | 76.5±1.2 | 0.52±0.03 | 0.57±0.05 | 0.82±0.02 | 0.48±0.04 | 0.75±0.03 | 0.28±0.05 |
| VAE | 81.3±1.5 | 0.62±0.04 | 0.66±0.06 | 0.86±0.02 | 0.57±0.04 | 0.87±0.02 | 0.51±0.04 |
| AT | 87.5±1.0 | 0.70±0.03 | 0.79±0.04 | 0.89±0.02 | 0.67±0.03 | 0.90±0.02 | 0.58±0.04 |
| GDN | 88.6±0.9 | 0.73±0.03 | 0.81±0.04 | 0.90±0.02 | 0.69±0.03 | 0.91±0.01 | 0.60±0.03 |
| CDA-ADN | 92.8±0.7 | 0.80±0.02 | 0.88±0.03 | 0.93±0.01 | 0.76±0.03 | 0.92±0.01 | 0.64±0.03 |

**Table 3. Cross-domain detection performance from WUSTL-IIoT-2021 (source) to ToN_IoT (target). Values are reported as mean±standard deviation over five independent runs.**

| Model | Acc (%) | MCC | Sen | Spe | F1 | AUC-ROC | AUC-PR |
|---|---|---|---|---|---|---|---|
| AE | 69.8±1.6 | 0.47±0.03 | 0.55±0.05 | 0.80±0.03 | 0.44±0.04 | 0.72±0.03 | 0.26±0.04 |
| VAE | 75.2±1.7 | 0.58±0.04 | 0.63±0.05 | 0.85±0.02 | 0.54±0.04 | 0.84±0.02 | 0.48±0.04 |
| AT | 81.0±1.3 | 0.66±0.03 | 0.75±0.04 | 0.88±0.02 | 0.62±0.03 | 0.88±0.02 | 0.55±0.03 |
| GDN | 82.1±1.2 | 0.69±0.03 | 0.77±0.04 | 0.89±0.02 | 0.64±0.03 | 0.89±0.02 | 0.57±0.03 |
| CDA-ADN | 88.2±0.9 | 0.77±0.02 | 0.86±0.03 | 0.92±0.01 | 0.73±0.03 | 0.91±0.01 | 0.61±0.03 |

For WUSTL-IIoT-2021→ACI-IoT-2023 (Table 2), CDA-ADN achieves the highest Accuracy (92.8±0.7%) and MCC (0.80±0.02), outperforming the strongest baselines GDN (88.6±0.9%, 0.73±0.03) and AT (87.5±1.0%, 0.70±0.03). The advantage is also reflected in Sensitivity (0.88±0.03) and F1-score (0.76±0.03), which confirms that CDA-ADN improves anomaly recall while maintaining balanced precision-recall behaviour. In addition, CDA-ADN yields the best threshold-independent metrics, with AUC-ROC of 0.92±0.01 and AUC-PR of 0.64±0.03, suggesting consistently stronger discrimination across decision thresholds under class imbalance.

For WUSTL-IIoT-2021 → ToN_IoT (Table 3), the overall performance of all methods decreases, which is expected due to a larger domain gap and more challenging target distribution. Nevertheless, CDA-ADN remains clearly superior, achieving the best Accuracy (88.2±0.9%) and MCC (0.77±0.02), with notable improvements over GDN (82.1±1.2%, 0.69±0.03) and AT (81.0±1.3%, 0.66±0.03). CDA-ADN also attains the highest Sensitivity (0.86±0.03) and F1-score (0.73±0.03), indicating better anomaly detection capability in the target domain while preserving strong Specificity (0.92±0.01). The AUC-ROC (0.91±0.01) and AUC-PR (0.61±0.03) further support that the proposed framework maintains robust ranking quality even when the transfer becomes more difficult.

Overall, the consistent gains across both transfer directions and across complementary metrics demonstrate that CDA-ADN improves cross-domain anomaly detection by jointly enhancing domain alignment and anomaly separability, yielding stable performance under varying degrees of distribution shift.

The improved performance of CDA-ADN can be attributed to its key components: input-output adaptation layers for domain alignment and contrastive learning for anomaly separation. These mechanisms enable CDA-ADN to achieve a high degree of generalization across different domains while maintaining consistent performance. As shown in Fig 3, CDA-ADN achieves the highest MCC and Sensitivity among all compared models, further confirming its superior anomaly detection capability.

## Ablation study

To evaluate the contributions of different components in CDA-ADN, we conduct an ablation study by systematically removing key modules and analyzing their impact on performance. Specifically, we assess the effects of removing the

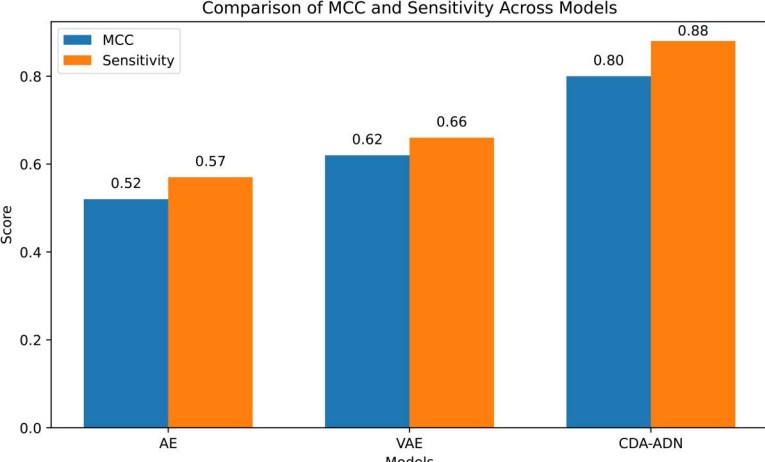

**Fig 3. MCC and sensitivity comparison across different models (mean±standard deviation).**

multi-granularity contrastive learning module, the dynamic input–output adaptation layers, and the conditional variational encoder, as well as a reduced configuration that removes both contrastive learning and adaptation layers. The results are presented in Table 4.

The results in Table 4 highlight that each component contributes meaningfully to cross-domain anomaly detection. Removing the multi-granularity contrastive learning module results in a clear drop in MCC from 0.80 to 0.73 and in sensitivity from 0.88 to 0.79. This suggests that contrastive learning enhances feature separability, making it easier to distinguish between normal and anomalous patterns under domain shift.

Removing adaptation layers also negatively impacts performance, as shown by the drop in MCC to 0.75 and sensitivity to 0.82. This confirms that the latent-guided input–output transformations effectively reduce cross-domain distribution mismatch and improve generalization to the target domain. Replacing the conditional variational encoder with a deterministic GRU autoencoder leads to a noticeable drop in MCC and sensitivity as well, indicating that probabilistic latent modeling together with KL regularization helps learn a more structured and transferable latent space.

When both contrastive learning and adaptation layers are removed, the performance deterioration is more pronounced. The MCC score drops further to 0.69, and sensitivity decreases to 0.75, indicating a substantial degradation in the model's ability to detect anomalies. The absence of both components causes the model to rely mainly on reconstruction learning, which is insufficient for capturing domain-invariant yet discriminative features. This confirms that the contrastive module and adaptation layers provide complementary benefits and work synergistically to enhance model robustness under domain shift.

**Table 4. Ablation study results on the target domain (mean±std).**

| Configuration | MCC | Sensitivity |
|---|---|---|
| Full Model (CDA-ADN) | 0.80±0.02 | 0.88±0.03 |
| Without Multi-Granularity Contrastive Learning | 0.73±0.03 | 0.79±0.04 |
| Without Adaptation Layers | 0.75±0.02 | 0.82±0.03 |
| Without Conditional Variational Encoder | 0.74±0.03 | 0.80±0.04 |
| Without Contrastive Learning and Adaptation Layers | 0.69±0.04 | 0.75±0.05 |

These findings underline the necessity of integrating conditional variational encoding, multi-granularity contrastive learning, and dynamic adaptation layers to achieve robust and generalizable anomaly detection. Removing any single component results in measurable degradation, and removing multiple components leads to a substantial decline in performance.

## Sensitivity analysis of hyperparameters

To further assess the robustness of CDA-ADN with respect to the contrastive learning objective, we perform a sensitivity analysis on two key hyperparameters: the balance coefficient $a$ between local and global contrastive terms, and the margin $m$ that controls the separation between positive and negative pairs in the contrastive loss. We vary $a$ and $m$ over a relatively wide range while keeping all other settings fixed, and report the classification accuracy on the target domain. The default configuration used in our main experiments is $\alpha = 0.5$ and $m = 0.5$.

As shown in Table 5, CDA-ADN maintains consistently high accuracy in a broad and practically reasonable range of $a$ and $m$. For $\alpha \in [0.2, 0.8]$ and $m \in [0.3, 0.8]$, the accuracy remains around 92%–93%, close to the default setting (92.8%), with only minor fluctuations. When the hyperparameters take more extreme values, such as $(\alpha, m) = (0.1, 0.1)$ or $(0.9, 1.0)$, the accuracy decreases more noticeably (to about 90.7%–90.9%), indicating that excessive emphasis on either local or global contrastive structure, or an overly small/large margin, can harm performance. Overall, these results demonstrate that CDA-ADN is robust to the choice of contrastive hyperparameters within a wide range around the default configuration.

In addition to contrastive-learning hyperparameters, we further examine two implementation-critical factors specific to CDA-ADN: the number $K$ of unlabeled normal target-domain samples used in the fine-tuning stage and the latent dimensionality $d_c$ of the conditional variational encoder. With $d_c = 64$ fixed, increasing $K$ from 10 to 50 and 100 improves the target-domain accuracy from about 90.8% to 92.0% and 92.8%, while further increasing to $K = 200$ yields only a marginal gain to roughly 93.0%, suggesting that performance saturates once a modest amount of normal target data is available. Conversely, with $K = 100$ fixed, varying $d_c$ in {16, 32, 64, 128} results in accuracies of approximately 91.3%, 92.3%, 92.8%, and 92.4%, respectively. These trends indicate that CDA-ADN already performs strongly with $K$ in the range of 50–100 and that $d_c$ in the range of 32–64 provides a stable trade-off between representation capacity and robustness.

## Conclusion

This paper presents the Cross-Domain Adaptive Anomaly Detection Network (CDA-ADN), an unsupervised transfer learning framework designed for multivariate anomaly detection in IoT traffic data. By integrating a variational sequence encoder with input-output adaptation layers and contrastive learning, the model effectively captures domain-invariant representations, reducing the need for labeled target domain data. Experimental evaluations on WUSTL-IIOT-2021 and ACI-IoT-2023 as well as ToN_IoT demonstrate that CDA-ADN outperforms traditional autoencoder-based methods in terms of accuracy, MCC, and sensitivity. The ablation study further validates the contributions of contrastive learning and adaptation layers in improving domain generalization. Future work will explore the extension of this approach to dynamic adaptation strategies for handling temporal variations in IoT environments.

**Table 5. Sensitivity of CDA-ADN to the balance coefficient $a$ and margin $m$ on the target domain (accuracy, mean±std over five runs).**

| $a$ | 0.2 | 0.5 | 0.5 | 0.8 | 0.8 | 0.1 | 0.9 |
|---|---|---|---|---|---|---|---|
| $m$ | 0.3 | 0.3 | 0.5 | 0.5 | 0.8 | 0.1 | 1.0 |
| Accuracy (%) | 92.1±0.8 | 92.6±0.7 | 92.8±0.7 | 92.4±0.8 | 92.0±0.9 | 90.9±1.1 | 90.7±1.2 |

 

## Author contributions

**Conceptualization:** Tiange Yuan, Di Zhai.

**Data curation:** Anchao Li.

**Investigation:** Tiange Yuan.

**Methodology:** Di Zhai.

**Software:** Anchao Li.

**Validation:** Di Zhai.

**Writing – original draft:** Tiange Yuan.

**Writing – review & editing:** Di Zhai, Anchao Li.

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
