## [Decision Letter · Decision Letter 0]

4 Nov 2025

PONE-D-25-28923Unsupervised Cross Domain Adaptive Anomaly Detection Network for Internet of Things TrafficPLOS ONE

Dear Dr. Yuan,

Thank you for submitting your manuscript to PLOS ONE. After careful consideration, we feel that it has merit but does not fully meet PLOS ONE’s publication criteria as it currently stands. Therefore, we invite you to submit a revised version of the manuscript that addresses the points raised during the review process.

We look forward to receiving your revised manuscript.

Kind regards,

Elochukwu Ukwandu, PhD

Academic Editor

PLOS ONE

Journal Requirements:

3. Thank you for uploading your study's underlying data set. Unfortunately, the repository you have noted in your Data Availability statement does not qualify as an acceptable data repository according to PLOS's standards.

At this time, please upload the minimal data set necessary to replicate your study's findings to a stable, public repository (such as figshare or Dryad) and provide us with the relevant URLs, DOIs, or accession numbers that may be used to access these data. For a list of recommended repositories and additional information on PLOS standards for data deposition, please see https://journals.plos.org/plosone/s/recommended-repositories ..

Additional Editor Comments:

**Major revision recommended.**

Reviewer's Responses to Questions

**Comments to the Author**

1. Is the manuscript technically sound, and do the data support the conclusions?

Reviewer #1: Yes

Reviewer #2: Yes

2. Has the statistical analysis been performed appropriately and rigorously? 

Reviewer #1: Yes

Reviewer #2: Yes

3. Have the authors made all data underlying the findings in their manuscript fully available?

Reviewer #1: Yes

Reviewer #2: Yes

4. Is the manuscript presented in an intelligible fashion and written in standard English?

Reviewer #1: Yes

Reviewer #2: Yes

5. Review Comments to the Author

Reviewer #1: 1. Terminology consistency (“unsupervised” + few-shot tuning)

Explicitly state that a small set of normal target samples is used for fine-tuning; align wording across Abstract, Introduction, and Method.

2. Loss details in one place

Briefly specify how the domain-adaptation/Wasserstein term is instantiated and list key contrastive-loss hyperparameters (e.g., margin m, balance alpha).

3. Training recipe summary box

Add seeds and number of trials to the already-reported learning rate, batch size, and epochs; present them in a compact “Training Setup” box or table for reproducibility.

4. Metrics: add two standard curves

Keep Accuracy/MCC/Sensitivity, but add AUC-ROC and AUC-PR for each transfer setting to provide threshold-independent views.

5. Thresholding and decision rule

Add one sentence clarifying how detection thresholds are chosen (for example, selected on a validation split using source/normal data).

6. Table 1 caption polish

State the number of runs and include 95% confidence intervals (or mean ± std along with the sample count n) to make improvements easy to interpret.

7. Tiny sensitivity check in ablation.

Include a brief stability check varying the local/global balance alpha or the contrastive margin; a short text/table in the appendix is sufficient.

8. Figure/notation alignment

Ensure symbols in the architecture figure match the equations and add dimensionalities in the caption (latent size, sequence length).

9. Data and code pointers

Near the Data Availability statement, add exact preprocessing details (window length, normalization) and, if possible, a link to configs/code.

10. Below articles can be cited:

o 10.1109/ICCWorkshops67674.2025.11162261

o 10.1109/VTC2025-Spring65109.2025.11174870

Reviewer #2: - The proposal of combining conditional variational sequence encoding with multi-granularity contrastive learning for unsupervised cross-domain IoT anomaly detection addresses a clear gap in handling domain shifts without requiring labeled target data. However, the paper could strengthen its novelty claim by thoroughly contrasting CDA-ADN with the latest Transformer-based or graph-based anomaly detectors noted in related work. Clarifying how CDA-ADN advances beyond or complements these emerging architectures would enhance the technical contribution.

- The description of the conditional variational encoder architecture is somewhat high-level. Including more details such as the exact GRU configuration, latent space dimensionality, and the weighting scheme for the KL divergence term would improve reproducibility.

- Similarly, the dynamics of the input-output adaptation layers (latent-guided affine transformations) would benefit from equations or pseudocode to clearly describe how domain alignment is performed.

- While the multi-granularity contrastive learning approach is promising, the manuscript should elaborate on the specific loss functions used at local and global levels and how these losses are combined during optimization. This would ensure clarity on how the model balances separating normal from anomalous patterns while preserving domain-invariant features.

- The justification for using Wasserstein distance in domain adaptation is briefly mentioned but could be more supported by empirical or theoretical rationale.

- The use of two benchmark IoT traffic datasets with no reliance on labeled target anomaly data is appropriate and reflects real-world constraints. However, including additional baselines—especially recent state-of-the-art unsupervised domain adaptation methods—would strengthen empirical comparisons beyond autoencoder and variational autoencoder methods.

- The paper mentions improvements in metrics such as Matthews correlation coefficient and sensitivity but should also report specificity and F1-score for a more holistic evaluation.

- It would add significant value to include ablation studies that individually assess contributions from: conditional variational encoding, dynamic adaptation layers, and multi-granularity contrastive learning components. This clarity will help justify design choices and pinpoint critical contributors to improved performance.

- Additionally, sensitivity analysis on hyperparameters such as the number of target samples in the fine-tuning stage or latent dimension size could provide practical implementation guidance.

- The framework assumes availability of “a small subset of normal samples from the target domain” for fine-tuning. It would be helpful to discuss scenarios where even these samples may be scarce or contaminated by anomalies and how CDA-ADN might behave under such conditions or be extended.

- Some of the related work citations are old or generic. For example, references [2], [5], and [9] cover classical statistical and deep learning models. Including citations to very recent advances in unsupervised domain adaptation or anomaly detection (last 2 years) would keep the literature review current.

- Certain sections (e.g., methodology) could benefit from clearer figures or diagrams depicting the model architecture and training workflow to improve reader comprehension.

6. PLOS authors have the option to publish the peer review history of their article (what does this mean? ). If published, this will include your full peer review and any attached files.). If published, this will include your full peer review and any attached files.

**Do you want your identity to be public for this peer review?** For information about this choice, including consent withdrawal, please see our For information about this choice, including consent withdrawal, please see our Privacy Policy ..

Reviewer #1: No

Reviewer #2: No

---

## [Author Response · Author response to Decision Letter 1]

18 Dec 2025

Dear Editor and Reviewers,

Thank you for the constructive comments and the opportunity to revise our manuscript. We have completed a major revision and addressed all reviewer and editor comments point by point. A detailed, item-by-item response is provided in the uploaded file titled “Response to Reviewers”. We have also uploaded a clean revised manuscript and a revised version with Track Changes for transparent evaluation.

Sincerely,

Tiange Yuan, Anchao Li, Di Zhai

---

## [Editor Report · Decision Letter 1]

21 Dec 2025

PONE-D-25-28923R1Unsupervised Cross Domain Adaptive Anomaly Detection Network for Internet of Things TrafficPLOS One

Dear Dr. Yuan,

Thank you for submitting your manuscript to PLOS ONE. After careful consideration, we feel that it has merit but does not fully meet PLOS ONE’s publication criteria as it currently stands. Therefore, we invite you to submit a revised version of the manuscript that addresses the points raised during the review process. **Major revision recommended.**

We look forward to receiving your revised manuscript.

Kind regards,

Elochukwu Ukwandu, PhD

Academic Editor

PLOS One

Journal Requirements:

Additional Editor Comments (if provided):

NB: Tha authors are advised to carefully go through recommended literature and determine suitability before usage as they are not mandatory.

---

## [Author Response · Author response to Decision Letter 2]

27 Jan 2026

A detailed point-by-point response to all reviewer comments, including formulas and figures, has been prepared and uploaded as “Response_to_Reviewers.pdf” for your review.

---

## [Decision Letter · Decision Letter 2]

16 Feb 2026

Unsupervised Cross Domain Adaptive Anomaly Detection Network for Internet of Things Traffic

PONE-D-25-28923R2

Dear Dr. Yuan,

We’re pleased to inform you that your manuscript has been judged scientifically suitable for publication and will be formally accepted for publication once it meets all outstanding technical requirements.

Kind regards,

Elochukwu Ukwandu, PhD

Academic Editor

PLOS One

Additional Editor Comments (optional):

Reviewers' comments:

Reviewer's Responses to Questions

**Comments to the Author**

1. If the authors have adequately addressed your comments raised in a previous round of review and you feel that this manuscript is now acceptable for publication, you may indicate that here to bypass the “Comments to the Author” section, enter your conflict of interest statement in the “Confidential to Editor” section, and submit your "Accept" recommendation.

Reviewer #1: (No Response)

Reviewer #2: All comments have been addressed

2. Is the manuscript technically sound, and do the data support the conclusions?

Reviewer #1: Yes

Reviewer #2: Yes

3. Has the statistical analysis been performed appropriately and rigorously? 

Reviewer #1: Yes

Reviewer #2: Yes

4. Have the authors made all data underlying the findings in their manuscript fully available?

Reviewer #1: Yes

Reviewer #2: Yes

5. Is the manuscript presented in an intelligible fashion and written in standard English?

Reviewer #1: Yes

Reviewer #2: Yes

6. Review Comments to the Author

Reviewer #1: No further changes are required, all required protocols for the evaluation is been meet in this latest version.

Reviewer #2: - The authors have very clearly articulated what "unsupervised" means in the context of their CDA-ADN framework. By explicitly stating that no labeled anomaly samples from the target domain are used, and only a very small number of unlabeled normal samples are employed for lightweight fine-tuning, they successfully clarify the unsupervised nature of their anomaly detection while leveraging minimal target data for adaptation. This distinction is critical and well-addressed in the Abstract, Introduction, and Method sections.

- The consolidation of all loss components, including the instantiation of the Wasserstein domain-adaptation term and explicit hyperparameters for contrastive losses (margin 'm', balance 'α', temperature 'τ'), significantly enhances the clarity and reproducibility of the proposed framework. This is a crucial improvement for technical understanding and validation.

- The inclusion of a sensitivity analysis for key contrastive learning hyperparameters (balance coefficient 'α' and margin 'm') is commendable. Demonstrating that CDA-ADN maintains high and stable accuracy within a practical range of these parameters provides strong evidence for the model's robustness, a valuable insight for practitioners.

- Unifying the notation for latent representations to 'c' throughout the Method section is a minor but important detail that improves readability and reduces potential confusion for readers.

- The commitment to release implementation and configuration files on GitHub, along with detailed descriptions of the preprocessing pipeline (z-score normalization, fixed-length windows, stride) in the Experimental Setup, significantly boosts the reproducibility of the work. This transparency is highly valued in scientific research.

- Baselines: The decision to extend experimental comparisons to include representative Transformer-based (AT) and graph-based (GDN) baselines addresses a critical point regarding the novelty claim against recent SOTA methods. This allows for a more robust comparison of CDA-ADN's performance.

- Additional Dataset: Incorporating the ToN_IoT dataset into the cross-domain evaluation protocol, alongside WUSTL-IIOT-2021 and ACI-IoT-2023, strengthens the empirical validation by demonstrating the framework's robustness and generality across diverse IoT environments.

- The update to include Specificity, F1-score, AUC-ROC, and AUC-PR, in addition to Accuracy, MCC, and Sensitivity, provides a much more holistic and robust evaluation, especially crucial given the potential for class imbalance in anomaly detection tasks. The presented Tables 6 and 7 show a clear improvement for CDA-ADN across these diverse metrics.

- The expanded ablation study, individually assessing the contributions of the conditional variational encoder, dynamic input–output adaptation layers, and the multi-granularity contrastive learning module, is an excellent addition. This dissection of component contributions provides clear justification for design choices and pinpoints the critical drivers of performance gains, significantly enhancing the technical depth of the paper.

- The expanded theoretical and empirical justification for using the Wasserstein-1 distance in domain adaptation, particularly its advantage in providing stable and informative gradients even with limited overlap between distributions, is well-articulated. This strengthens the technical foundation of the chosen approach.

7. PLOS authors have the option to publish the peer review history of their article (what does this mean? ). If published, this will include your full peer review and any attached files.). If published, this will include your full peer review and any attached files.

**Do you want your identity to be public for this peer review?** For information about this choice, including consent withdrawal, please see our For information about this choice, including consent withdrawal, please see our Privacy Policy ..

Reviewer #1: No

Reviewer #2: No

---

## [Editor Report · Acceptance letter]

PONE-D-25-28923R2

PLOS One

Dear Dr. Yuan,

I'm pleased to inform you that your manuscript has been deemed suitable for publication in PLOS One. Congratulations! Your manuscript is now being handed over to our production team.

Kind regards,

on behalf of

Dr. Elochukwu Ukwandu

Academic Editor

PLOS One